# OpenReview forum: "Diagnose, Localize, Align: A Full-Stack Framework for Reliable LLM Multi-Agent Systems under Instruction Conflicts"
_ICLR.cc/2026/Conference — ICLR 2026 Conference Withdrawn Submission_

### Official Review · Reviewer_Y2Xc · 2025-10-14

**Soundness:** 1
**Presentation:** 2
**Contribution:** 2
**Rating:** 2
**Confidence:** 4

**Summary:**

The paper presents a complete framework for diagnosing conflicting commands in multi-agent systems, locating the key part of the model that resolves these conflicts, and fixing it through an efficient "surgical" fine-tuning method that makes the entire system more reliable in the face of complex and contradictory commands while avoiding costly and time-consuming retraining of the entire model. This makes the entire system more reliable in the face of complex and contradictory instructions.

**Strengths:**

- Multi-agent system diagnosis is a significant problem when applied to real-world scenarios. In this paper, the author proposes a three-step solution for identifying, locating, and improving LLMs for a multi-agent system, making it a meaningful candidate for publication in the conference.
- The author's experiment does not involve any closed-weight language model, making the result completely reproducible.
- Introducing a new metric called CRAC for evaluating the multi-agent system trajectories, which brings interpretability to the multi-agent system diagnosis.

**Weaknesses:**

- Lacking implementation details. Please see the questions.
- The author proposed an LLM-based evaluation metric called CRAS for the preference evaluation, but there is no ablation study on CRAS to show the correlation between CRAS and human evaluation, only the correlation between CRAS and downstream performance.
- In `Section 3.2`, the author builds a parameter localization method upon CRAS by comparing the CRAS difference between a non-conflicted and a conflicted prompt (`Equation (8)`). But according to `Figure 2c`, the author shows that the CRAS is positively correlated with the downstream performance. This raises the question about the necessity of CRAS: why can't we just sample a few examples using two different prompts and use their average performance instead of CRAS? It's unclear how many benefits CRAS brings to the localization and the following SAIL. Therefore, the author should add a comparison between using CRAS and the average accuracy.
- Based on the localization method introduced in `Section 3.2`, the author further proposes a surgical fine-tuning method by adding a LoRA layer to the located conflicting transformer layer. However, the best performance of LoRA has been demonstrated by many works that happen when adding LoRA layers to all transformer layers. In this paper, there are no experiments of: (1) fine-tuning all layers with LoRA and the author's SAIL method, (2) full parameter SFT for the selected layer, and (3) the cost/learning dynamics/training time between these methods. Lacking these experiments significantly decreases the soundness of this paper.

**Questions:**

- What is the backbone model you used for CARS evaluation, ie, the model parameterized by $\theta_{\text{gen}}$?
- What are `Figure 7: model CRAS sensitivity to learning rate` and `Figure 8: Model CRAS sensitivity to LoRA rank` for? Does the CRAS model also evolve during training?
- For the paper arrangement, I would suggest reducing the use of bolding and putting Figure 6 beside Figure 9 for better readability.

---

### Official Review · Reviewer_MSs9 · 2025-11-01

**Soundness:** 1
**Presentation:** 3
**Contribution:** 2
**Rating:** 2
**Confidence:** 2

**Summary:**

This paper presents DLA to improve the reliability and alignment of MAS especially under instruction hierarchies where system level directives may override or conflict with agent level goals.

**Strengths:**

1. Clarity: the pipeline is clear and key metrics/quantities are formally defined. It’s well-written and easy to understand.
2. Significance: The results shows consistent CRAS gains and frequent ACC gains across different backbones and MAS frameworks.

**Weaknesses:**

1. Evaluation: While the paper reports ACC on MMLU, SciBench, GPQA, and MedQA to demonstrate that SAIL does not harm general capability, these benchmarks are not designed to measure hierarchical or role-conflict reasoning. Thus, improvements in CRAS primarily reflect optimization toward the internal rubric rather than validated gains on real hierarchical tasks. ACC only confirms the model’s base competence remains decent, not that the claimed alignment transfers beyond CRAS. I would recommend authors to check related benchmarks not under the context of MAS as a reference, such as [1] [2] to see what other metrics can be used to show the success of your method. This seems to be a very fundamental flaw of this paper.
2. LLM as a judge for CRAS can be bittle: the rubrics in CRAS are prompt-based in nature. And there are many prior works shows that these types of methods can be biased [3,4].
3. Novelty: The alignment loss is explicitly token-weighted DPO-style with LoRA adapters, i.e., an adaptation of DPO and LoRA rather than a novel training objective or adapter mechanism. For conflict sensitivity analysis, there are potentially concurrent works such as [5] and it’s good to have some discussion can compare the conclusion with the one in the paper.

[1] Geng, Yilin, et al. "Control illusion: The failure of instruction hierarchies in large language models." arXiv preprint arXiv:2502.15851 (2025).

[2] Zhang, Zhihan, et al. "IHEval: Evaluating language models on following the instruction hierarchy." arXiv preprint arXiv:2502.08745 (2025).

[3] Li, Songze, et al. "LLMs Cannot Reliably Judge (Yet?): A Comprehensive Assessment on the Robustness of LLM-as-a-Judge." arXiv preprint arXiv:2506.09443 (2025).

[4] Li, Haitao, et al. "Llms-as-judges: a comprehensive survey on llm-based evaluation methods." arXiv preprint arXiv:2412.05579 (2024).

[5] Zeng, Siqi. "Dissecting Role Conflicts in Instruction Following." Mechanistic Interpretability Workshop at NeurIPS 2025.

**Questions:**

See weaknesses above, and 1 more question:
1. The conflict datasets in D.1 cover seven synthetic categories. Could you clarify how representative these conflicts are of real multi-agent innteraction and what are the principles to choose them? Do you plan to evaluate on the performance on held out conflict types, such as conflicting tool actions when two agents both attempt to write to or modify the same shared file overwriting each other’s output yet these agents have different hierarchical roles?

---

### Official Review · Reviewer_kYLX · 2025-11-01

**Soundness:** 3
**Presentation:** 3
**Contribution:** 3
**Rating:** 4
**Confidence:** 3

**Summary:**

This paper proposes a three-stage (Diagnose-Localize-Align) framework for enhancing reliability in LLM-based multi-agent systems (MAS) under instruction conflicts, introducing the Contextualized Role Adherence Score (CRAS) for diagnosis, identifying conflict-sensitive middle layers via attention drift analysis, and developing Surgical Alignment of Instruction Layers (SAIL) for targeted optimization. It validates the framework across benchmarks (e.g., MedQA, SciBench) and MAS frameworks (e.g., AutoGen), showing improved instruction hierarchy compliance without full-model finetuning.

**Strengths:**

1. Originality: Combines diagnostic scoring, attention-based localization, and targeted alignment into a cohesive pipeline for MAS instruction conflict resolution.
2. Quality: Validates results across diverse backbones, datasets, and MAS frameworks, ensuring generalizability.
3. Clarity: Structures the three-stage framework logically, with clear definitions of key components (CRAS dimensions, SAIL’s LoRA deployment).
4. Significance: Addresses a real-world barrier to reliable MAS deployment, providing actionable tools (CRAS, SAIL) for practitioners.

**Weaknesses:**

1. Relies heavily on existing techniques with incremental adjustments, lacking breakthrough innovations in alignment or localization.
2. Ablation studies for reward mechanisms (Table 2) do not analyze the patterns or reasons that "Constant Reward" underperforms beyond surface observations.
3. The CRAS rubric’s programmatic generation lacks step-by-step transparency, making replication challenging.
4. Fails to test the framework on long-horizon or real-world MAS tasks, limiting evidence of practical scalability.
5. Provides no quantitative analysis of how frequently "instruction conflicts" occur in real MAS deployments, raising doubts about the problem’s prevalence and the framework’s broader applicability.

**Questions:**

Address the weaknesses.

---

### Official Review · Reviewer_tS9D · 2025-11-01

**Soundness:** 2
**Presentation:** 2
**Contribution:** 2
**Rating:** 6
**Confidence:** 2

**Summary:**

This paper introduces a comprehensive three-stage framework ("Diagnose, Localize, Align") designed to enhance the reliability of LLM-based multi-agent systems (MAS), with a specific focus on resolving instruction conflicts that emerge between system-level and user-level directives. The key contributions include the Contextualized Role Adherence Score (CRAS), a context-aware and multi-dimensional metric for evaluating agents' adherence to role and instruction hierarchies; an attention-head/layer localization analysis that identifies the internal model regions responsible for instruction arbitration under conflict; and Surgical Alignment of Instruction Layers (SAIL), a parameter-efficient fine-tuning method that integrates LoRA adapters exclusively into attention focal layers identified through attention drift analysis, optimizing a weighted focal-head preference loss.

**Strengths:**

1. The paper addresses an important problem in LLM-based multi-agent systems — role adherence under instruction conflicts — which extends beyond traditional single-agent instruction following, offering both practical relevance and academic value.
2. The proposed method is validated across multiple benchmarks (MMLU, SciBench, GPQA, MedQA) and MAS frameworks (Dylan, MacNet, AutoGen, SelfConsistency), with comprehensive experimental comparisons.

**Weaknesses:**

1. Although CRAS employs a well-defined rubric and prompting strategy, it still relies on another LLM evaluator for scoring, which may introduce evaluation bias or inconsistency.
2. Artificial definition of “conflict”:The conflict dataset is generated via templates covering seven conflict types. While systematic, it may not fully capture the complexity of real-world multi-turn dialogues, limiting generalization.
3. Lack of theoretical rigor: Despite formal notation, the paper lacks theoretical guarantees — e.g., there is no formal proof of SAIL’s optimality or convergence, nor justification for why focal-layer adaptation does not compromise global model capacity.
4. Limited coverage of dynamic or long-horizon interactions: The work primarily focuses on static roles and single-turn conflicts, without addressing multi-turn dialogues, dynamic role switching, or long-term memory effects.

**Questions:**

1. Since CRAS evaluation fully depends on another LLM as the scorer, have you conducted human evaluation or cross-model validation to verify its consistency and reliability?
2. Your attention drift analysis suggests that instruction arbitration predominantly occurs in mid layers, which is an intriguing finding. Do you have any theoretical explanation or prior hypothesis supporting this phenomenon?
3. Given that CRAS depends on a rubric and prompt design, to what extent is it sensitive to prompt variation? Have you tested CRAS stability across different prompt formulations?

---

### Note · Authors · 2025-11-23

I have read and agree with the venue's withdrawal policy on behalf of myself and my co-authors.